# Oncological and Peri-Operative Outcomes of Percutaneous Cryoablation of Renal Cell Carcinoma for Patients with Hereditary RCC Diseases—An Analysis of European Multi-Centre Prospective EuRECA Registry

**DOI:** 10.3390/cancers15133322

**Published:** 2023-06-24

**Authors:** Filzah Hanis Osman, Vinson Wai-Shun Chan, David J. Breen, Alexander King, Tommy Kjærgaard Nielsen, Julien Garnon, Des Alcorn, Brunolf Lagerveld, Ole Graumann, Francis Xavier Keeley, Miles Walkden, Éric de Kerviler, Tze Min Wah

**Affiliations:** 1School of Medicine, Faculty of Medicine and Health, University of Leeds, Leeds LS2 9JT, UK; 2Leeds Institute of Medical Research, University of Leeds, Leeds LS2 9JT, UK; 3Royal Derby Hospital, University Hospitals of Derby and Burton NHS Foundation Trust, Derby DE22 3NE, UK; 4Division of Surgery and Interventional Science, University College London, London WC1E 6BT, UK; 5Department of Radiology, Southampton University Hospitals, Southampton SO16 6YD, UK; 6Department of Urology, Aarhus University Hospital, 8200 Aarhus, Denmark; 7Department of Interventional Radiology, Nouvel Hôpital Civil, 1 Place de l’Hôpital, 67000 Strasbourg, France; 8Department of Interventional Radiology, Gartnavel General Hospital, Glasgow G12 0YN, UK; 9Department of Urology, OLVG, Amsterdam 1091 AC, The Netherlands; 10Department of Radiology, Odense University Hospital, 5000 Odense, Denmark; 11Bristol Urological Institute, North Bristol NHS Trust, Bristol BS10 5NB, UK; 12Department of Imaging, University College London Hospitals NHS Foundation Trust, London WC1N 3BG, UK; 13Radiology Department, Saint-Louis Hospital, AP-HP, 1 Avenue Claude-Vellefaux, 75475 Paris, CEDEX 10, France; 14Department of Diagnostic and Interventional Radiology, Institute of Oncology, St. James’s University Hospital, Leeds Teaching Hospitals NHS Trust, Leeds LS9 7TF, UK

**Keywords:** renal cell carcinoma, inherited RCC, hereditary RCC, percutaneous cryoablation, image-guided cryoablation, small renal masses, Von-Hipple Lindau disease

## Abstract

**Simple Summary:**

Inherited renal cell carcinoma (RCC) typically presents earlier in life with multifocal and bilateral tumours. Treatment of such tumours is often challenging due to their bilateral presentation and high risk of recurrence or development of new disease. Therefore, the goal of treatment is to achieve oncological control while preserving renal function as much as possible. The purpose of this study was to evaluate the safety, efficacy, and preservation of renal function of percutaneous cryoablation (PCA) for small renal masses (SRM) in inherited RCC. We reviewed European data and found that image-guided cryoablation is safe, maintains good kidney function, and is effective in controlling cancer arising from hereditary small kidney cancers.

**Abstract:**

This study aims to evaluate the safety, efficacy, and renal function preservation of percutaneous cryoablation (PCA) for small renal masses (SRMs) in inherited RCC syndromes. Patients with inherited T1N0M0 RCCs (<7 cm) undergoing PCA from 2015 to 2021 were identified from the European Registry for Renal Cryoablation (EuRECA). The primary outcome was local recurrence-free survival (LRFS). The secondary outcomes included technical success, peri-operative outcomes, and other oncological outcomes estimated using the Kaplan–Meier method. Simple proportions, chi-squared tests, and *t*-tests were used to analyse the peri-operative outcomes. A total of 68 sessions of PCA were performed in 53 patients with RCC and 85 tumours were followed-up for a mean duration of 30.4 months (SD ± 22.0). The overall technical success rate was 99%. The major post-operative complication rate was 1.7%. In total, 7.4% (2/27) of patients had >25% reduction in renal function. All oncological events were observed in VHL patients. Estimated 5-year LRFS, metastasis-free survival, cancer-specific survival, and overall survival were 96.0% (95% CI 75–99%), 96.4% (95% CI 77–99%), 90.9% (95% CI 51–99%), and 90.9% (95% CI 51–99%), respectively. PCA of RCCs for patients with hereditary RCC SRMs appears to be safe, offers low complication rates, preserves renal function, and achieves good oncological outcomes.

## 1. Introduction

Renal cell carcinoma (RCC) occurs in both sporadic and heritable forms, with hereditary RCC accounting for 2–4% of cases of RCC [1]. Whilst sporadic RCC typically presents as a solitary lesion commonly in older patients beyond 60 years of age, RCC arising due to inherited diseases presents earlier in life with multifocal and bilateral tumours [2].

The four major inherited RCC diseases associated with increased risk of RCCs are Von Hippel-Lindau disease (VHL), hereditary leiomyomatosis and renal cell cancer (HLRCC), hereditary papillary renal carcinoma (HPRC), and Birt-Hogg-Dube syndrome (BHD). These autosomal dominant susceptibility syndromes arise due to gene mutations in *VHL*, fumarate hydratase (FH), *MET*, and folliculin (FLCN), respectively [3].

Treatment of inherited RCC is often challenging due to the bilateral nature of tumours and the high risk of recurrence or development of new disease. Management of such tumours primarily focuses on achieving good oncological durability and preventing metastatic disease without compromising renal function in the long-term, therefore avoiding the development of end-stage renal disease. As such, nephron-sparing treatment is preferred in these patients. In the last decade, partial nephrectomy (PN) has replaced radical nephrectomy (RN) as the standard of care for small renal masses of stage T1a, mainly due to its ability to maintain good oncological outcomes whilst preserving renal function [4,5]. However, repeated PN in multifocal and recurrent RCCs can be technically challenging. Patients are exposed to a risk of major complications and reoperation as high as 20%, with just under 4% of patients requiring long-term haemodialysis [6,7].

In recent years, image-guided ablation (IGA) employing heat-based energy, such as radiofrequency ablation (RFA) and microwave ablation, or cold-based energy, such as cryoablation (CRYO), has gained attention for its ability to achieve comparable oncological outcomes to PN and better renal function preservation [4]. However, data are limited due to low certainty of evidence [4]. In the European Association of Urology (EAU)’s latest guidelines, RFA and CRYO have been recommended as alternative treatment options for frail or comorbid patients who are deemed unfit for surgery [8].

As expected, IGA has been increasingly adopted as the preferred treatment option in patients with hereditary RCC syndromes due to the need to maintain good oncological durability whilst preserving renal function. Many studies have supported this approach, with positive findings, good oncological outcomes, and long-term preservation of renal function in IGA patients with hereditary RCC syndromes [5,9,10]. However, it is difficult to obtain large-scale data on the management of RCCs in hereditary diseases, with most of the current literature focusing on VHL patients. Hence, this study, which utilizes a multi-centre, European, prospectively-maintained database, aims to report a large-scale cohort analysis of patients with various inherited RCC syndromes treated with percutaneous cryoablation (PCA) across major European centres.

## 2. Materials and Methods

Institutional review board approval and patient consent were not required for this registry-based retrospective study. Patients with inherited RCC syndromes with localised cT1aN0M0 or cT1bN0M0 treated with PCA at 11 European centres between 2015 and 2021 were identified from the European Registry for Renal Cryoablation (EuRECA), a prospectively-maintained, multi-centre database. cT1a and cT1b renal masses were defined as having maximum tumour diameters of ≤4 cm and >4 cm and ≤7 cm, respectively, on radiological imaging based on the American Joint Committee on Cancer (AJCC) staging manual [11].

Patient age, sex, race, comorbidities (Charlson Comorbidity Index (CCI)), and clinical history were analysed. Tumour characteristics such as size and complexity were measured via maximum diameter and with the components of the R.E.N.A.L. nephrometry score, respectively. Patient baseline eGFR values were collected as baseline pre-operative renal function.

CRYO was performed with a percutaneous approach under image guidance as per standard institutional protocol. Primary technical success was defined as complete treatment response after one treatment session, with no evidence of residual disease at the zone of ablation after treatment; overall technical success referred to complete treatment response regardless of the number of treatment sessions. Patients were followed-up after treatment with CRYO according to standard institutional protocol. Oncological outcomes such as local recurrence-free survival (LRFS), metastasis-free survival (MFS), cancer-specific survival (CSS), and overall survival (OS) were evaluated from the time of treatment to the time of event using Kaplan–Meier curves. Local recurrent disease was defined as any evidence of enhancement within the zone of ablation during follow-up imaging after negative initial post-CRYO imaging. Post-operative complications were classified using the Clavien-Dindo (CD) scale, which consists of four severity grades (grades I, II, III, and IV) [12]. Complications were considered ‘minor’ if they were CD grades I and II (complications that may or may not require pharmacological management) and ‘major’ if they were CD grades III (those requiring surgical, radiological, or endoscopic intervention) or IV (life-threatening complications). Survival rates and corresponding 95% confidence intervals (95% CI) were reported. The chi-squared test and two sample *t*-test were used to determine any associations between complications and reduction (>10% and >25%) in renal function with patient and tumour characteristics, such as age, sex, type of inherited RCC disease, pre-operative renal function, and R.E.N.A.L nephrometry score. Statistical analyses were performed using Stata 17 (Stata Corp, College Station, TX, USA). Descriptive statistics including mean, SD, median, and IQR are reported.

## 3. Results

### 3.1. Baseline Characteristic of Included Patients

Fifty-three patients with inherited RCC syndromes from 11 major academic centres (Table A1) across Europe were treated with CT- or MRI-guided PCA from 2015 to 2021. The baseline characteristics of the included patients are outlined in Table 1. Forty-one patients had VHL disease, while one, two, and nine patients had HLRCC, HRPC, and BHD, respectively. Thirty-four percent presented with a known family history of renal cancer. Three-quarters (75.5%) of the included patients presented between the ages of 23 to 59 years. Nine patients had solitary kidneys, among whom eight had undergone contralateral RN due to previous RCCs.

Several patients had previously undergone multiple treatments for prior RCCs on either one of the kidneys, as outlined in Table 2. On the same kidney, 15 out of 53 patients had undergone PN for previous RCCs, 14 had previously been treated with PCA, and seven had undergone RFA. On the contralateral kidney, 8 had undergone RN, 10 had undergone PN, 12 had PCA, 3 had RFA, and 2 had irreversible electroporation (IRE).

Nineteen patients presented with multiple tumours over the course of the study, amounting to a total of 85 tumours treated with PCA over 68 sessions. The mean number of treated RCCs per patient was 1.60 (SD ± 1.04), with a mean tumour size of 2.5 cm (SD ± 1.0) and mean R.E.N.A.L. nephrometry score of 6.9 (SD ± 1.9). Eleven patients had developed subsequent de novo RCCs, with a mean disease-free period of 13 months (IQR: 0.67–47).

### 3.2. Treatment Efficacy, Peri-Operative Complications, and Change in eGFR

In 75 tumours with available follow-up data, 94.7% (71/75, 95% CI 87–99%) achieved primary technical success, while overall technical success was achieved in 99% (77/78, 95% CI 93–100%) of tumours with available consecutive follow-up data. In sessions with available peri-operative data, none had intra-operative complications (0/64), while 6.9% (4/58, 95% CI 3.3–16.7%) had post-operative complications. Post-operatively, two VHL patients experienced minor CD-I complications, and one VHL patient experienced a CD-II complication. Only one VHL patient experienced a major complication (CD-III), which was clot colic requiring ureteric stenting under general anaesthesia, suggesting a major complication rate of 1.7% (1/58, 95% CI 1.7–9.2%).

Pre-operatively, baseline eGFR was generally lower (*p* = 0.067) in patients with solitary kidneys (61.76 mL/min/1.73 m^2^, SD ± 18.63) as compared to those with two functioning kidneys (93.2 mL/min/1.73 m^2^, SD ± 46.44). In 27 patients who had undergone 35 treatment sessions with available pre- and post-operative eGFR data, the mean post-operative decrease in eGFR was 6.02 mL/min/1.73 m^2^ (SD ± 17.24) (mean −5.3% change, SD ± 16.8). Ten patients (37%, 95% CI 9.3–57.6%) in 11 treatment sessions developed more than 10% post-operative reduction in renal function. Two patients (7.4%, 95% CI 5.0–24.3%) in two sessions developed more than 25% post-operative reduction in renal function. Two patients who had solitary kidneys out of the 27 patients with available peri-operative data did not experience a significantly different change in renal function as compared to the rest of the patients (*p* = 0.893).

Chi-squared tests and *t*-tests were performed to assess the association between patient and tumour characteristics in relation to >10% (Appendix: Table A2) and >25% post-operative reduction in eGFR (Table 3) and post-operative complications No characteristics were associated with post-operative complications. Pre-operative eGFR (*p* = 0.001) and hypertension (*p* = 0.038) were significantly associated with >25% reduction eGFR. The two patients who developed >25% reduction in eGFR had a higher mean pre-operative eGFR of 143.78 mL/min/1.73 m^2^ (SD ± 6.04) compared to the rest of the patients (pre-operative eGFR of 75.65 mL/min/1.73 m^2^, SD ± 4.84). The association between pre-operative eGFR and >10% reduction in eGFR is displayed in Appendix A Figure A1. No other factors were found to be significantly associated. This was likely due to the small sample size and paucity of events seen in the cohort.

### 3.3. Oncological Durability

Patients were followed up over a mean duration of 30.4 months (SD ± 22.2). All observed oncological events were in VHL patients. One local recurrence was observed in a VHL patient at 22 months since the first treatment, which required repeated PCA. The recurrence was observed in the patient’s solitary left kidney (previous nephrectomy of their right kidney performed due to RCC) and had a RENAL nephrometry score of 9. The solid tumour measured 33 mm in diameter and was <50% exophytic. The anteriorly located lesion crossed the polar line for more than 50%, was less than 4 mm from the collecting system, and touched the renal artery.

The results suggested a 5-year LRFS rate of 96.0% (95% CI 75–99%). The 5-year MFS was 96.4% (95% CI 77–99%) due to one metastasis occurring at 60 months following initial treatment. The CSS and OS rates were both 90.9% (95% CI 51–99%), with only one death of a VHL patient, which was RCC-related, occurring at 42.3 months since the first treatment. The 1-year survival rates for LRFS, MFS, CSS, and OS were all 100%, and the 3-year survival rates were 96% (95% CI 75–99%), 96.4% (95% CI 77–99%), 100%, and 100%, respectively. The Kaplan–Meier curves for LRFS, MFS, CSS, and OS are shown in Figure 1.

## 4. Discussion

Over the past few decades, the management of hereditary RCCs has evolved from radical nephrectomy to minimally-invasive partial nephrectomy and more recently, IGA [13]. The paradigm shift towards the use of minimally-invasive treatment in patients with hereditary RCC in recent years has led to a significant improvement in the prognosis of these patients [13]. Despite the progress made in achieving better outcomes for this group of patients, there is still a lack of international guidance on the use of surgical treatment or IGA in managing SRMs in hereditary RCC, primarily due to the lack of prospective, large cohort studies especially related to CRYO [8]. Additionally, most of the existing small RCC cohort studies do not include patients with inherited diseases [14,15,16].

This is the first large-scale study to report the outcomes of PCA in patients with various inherited RCC syndromes in high-volume centres across Europe, based on the EuRECA registry. This study included patients with four major hereditary RCC diseases; most commonly VHL and BHD, followed by less common HLRCC and HRPC. One-third of the patients were treated for multiple tumours over the course of the study, and a significant proportion had undergone previous treatment for RCC on the same (43.4%) and contralateral (54.7%) kidneys. Development of new de novo RCCs was also observed in 20% of the cohort, where the mean time to detection of the new de novo RCC was 13 months (IQR: 0.67–47). These findings substantiate the fact that many patients with inherited RCC present in a complex manner with multiple, recurring tumours, often coupled with diminished renal function due to earlier treatment for previous tumours. This reiterates the importance of adopting a minimally invasive approach to provide local oncological control without compromising renal function.

In the experience of 11 major institutions in the EuRECA registry, PCA has proven to be safe in managing inherited RCC, with no intra-operative complications and few post-operative complications observed. To date, only a few studies have reported on the safety and efficacy of PCA of renal tumours in hereditary RCC [17,18]. Our findings are similar to those of Chan et al., whose cohort of 17 VHL patients undergoing multi-modal IGA (RFA, CRYO, and IRE) experienced one major CD-III complication in 50 treatment sessions, as well as a one CD-I and eight CD-II complications [9]. Only one study reported on the peri-operative outcomes of PCA in BHD patients, which had promising results with few major CD-III complications observed [17]. To our knowledge, the safety and oncological durability of PCA in HLRCC and HPRC patients have not yet been investigated. Our experience with 68 sessions of PCA in patients with hereditary RCC represents the largest cohort yet, with a post-operative major complication rate of 1.7%, similar to that of a non-hereditary-RCC cohort [5].

In terms of post-operative change in renal function, patients in our cohort experienced minimal reduction in renal function with an average 5.3% change (SD ± 16.8). One-fifth (6 out of 27) had improved renal function post-PCA, while only 7.4% (2 out of 27) developed >25% reduction of eGFR post-PCA. While these findings are concordant with those of Buy et al. who reported unchanged or improved renal function post-PCA in a mixed cohort of sporadic and hereditary RCC [19], they are to be interpreted with caution due to natural physiological variations in renal function as well as the relative unreliability of eGFR. Based on the current literature, PCA is known to provide good preservation of renal function, even in solitary kidneys [19,20]. Therefore, our findings suggest that PCA in hereditary RCC could be just as effective as in sporadic RCC in terms of renal function preservation.

In 53 patients with a mean follow up of 30.4 months, the estimated 5-year oncological durability for LRFS, MFS, CSS, and OS were 96.0%, 96.4%, 90.9%, and 90.90%, respectively, with no BHD, HLRCC, and HRPC patients experiencing any oncological events. These results reflect the potential less aggressiveness of these diseases, albeit the small sample size and theoretical genetical aggressiveness of HLRCC. Our 5-year LRFS was similar to that in the existing literature for multi-modal IGA in VHL patients [9] as well as for PCA in cT1 RCC in a non-hereditary RCC cohort [21]. To our knowledge, there are no other cohort studies that have reported oncological outcomes of PCA in patients with hereditary RCC.

In comparison to repeated PN, the most recent series of salvage PN reported by Bratslavsky et al. in 2008 showed a major complication rate as high as 46%, with 23% of patients ultimately requiring radical nephrectomy [7]. This suggests a superior peri-operative profile and at least a non-inferior overall profile of PCA for hereditary small RCCs, despite technological advancements in recent years.

In addition to the genetic causes outlined above, visceral adipose tissue is also a known risk factor for the development of RCC [22]. Further studies to evaluate the amount of adipose tissue in patients affected by hereditary RCC diseases can facilitate a better understanding of the pathogenesis and the relationship with adipose tissues in these hereditary syndromes.

Despite being the first large-scale registry-based study of PCA in patients with hereditary RCC, this study does not come without its limitations. Although the EuRECA registry represents the largest sample size of patients with hereditary RCC yet, the sample size still remains too small to achieve statistical power. We were unable to determine any associations between patient demographics and peri-operative outcomes due to the lack of statistical power. Furthermore, comparison of outcomes between different inherited RCC groups was not possible due to large discrepancies in the numbers of patients with each of the inherited RCC diseases. Finally, it is worth noting this manuscript only included cryoablation as a modality and further modalities such as radiofrequency ablation could be explored in patients with both sporadic and hereditary RCC [23].

## 5. Conclusions

This registry-based study has found PCA to be a feasible and safe option in the management of small hereditary RCCs due to its ability to provide oncological durability whilst preserving long-term renal function. In our cohort of patients, PCA was associated with good peri-operative outcomes, with no intra-operative complications and few post-operative complications observed. This study, however, is limited by low statistical power due to the rare nature of hereditary RCC. More large-scale studies are needed to confirm the efficacy and safety of PCA in each of the various hereditary diseases, especially in the less common diseases such as HLRCC and HPRC.

## Figures and Tables

**Figure 1 cancers-15-03322-f001:**
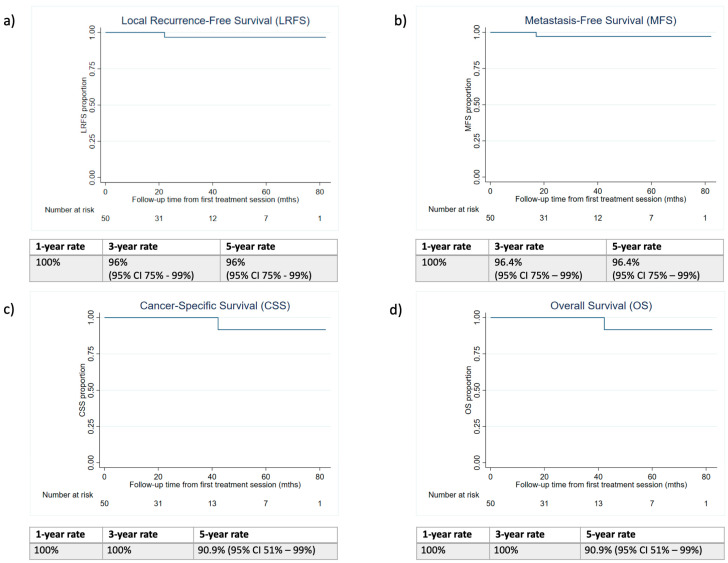
Kaplan–Meier curves for (**a**) Local Recurrence-Free Survival (LRFS); (**b**) Metastasis-Free Survival (MFS); (**c**) Cancer-Specific Survival (CSS); (**d**) Overall Survival (OS).

**Table 1 cancers-15-03322-t001:** Patient demographics and tumour characteristics.

No. of Patients (*n* = 53)		
**Variable**	**Frequency**	**%**
Age (years)		
<30	1	1.9
30–39	15	28.3
40–49	14	26.4
50–59	10	18.9
60–69	9	17.0
70–79	3	5.7
>80	1	1.9
Sex		
Male	30	56.5
Female	23	43.4
Race		
Caucasian	52	98.1
Asian	1	1.9
Type of hereditary disease		
VHL	41	77.4
HLRCC	1	1.9
HPRC	2	3.8
BHD	9	17.0
Family history of renal cancer		
Unknown	26	49.0
Yes	18	34.0
No	9	17.0
Solitary kidney		
No	44	83
Yes	9	17
	**Mean**	**SD**
No. of tumours per patient	1.6	1.0
Charlson Comorbidity Index	2.0	1.9
Baseline eGFR (mL/min/1.73 m^2^)	88.4	44.7
Follow-up duration (months)	30.4	22.0
No. of tumours (n = 85)		
	**Frequency**	**%**
Laterality		
Right	42	49.4
Left	43	50.6
	**Mean**	**SD**
Size of tumour (cm)	2.46	1.0
R.E.N.A.L. nephrometry score	6.9	1.9

**Table 2 cancers-15-03322-t002:** Breakdown of patients who had undergone prior treatment for RCCs.

Previous Treatment for RCCs on Same Kidney (*n* = 26)		Previous Treatment for RCCs on Contralateral Kidney (*n* = 29)	
Treatment	No. of Patients, *n*	Treatment	No. of Patients, *n*
PN only	8	PN only	6
PN + PCA	5	PN + PCA	2
PN + RFA	1	PN + RFA	1
PN + RFA + PCA	1	PN + IRE	1
PCA only	6	PCA only	7
PCA + RFA	2	PCA + RFA	2
RFA only	3	PCA + IRE	1
		RN only	8
		Unknown	1

**Table 3 cancers-15-03322-t003:** Factors associated with >25% reduction in eGFR.

**Variable**		**>25% Reduction in eGFR**	**Mean**	**SD**	** *p* ** **-Value (*t*-Test)**
Renal Nephrometry Score		Yes	9.5	0.5	0.071
		No	6.92	0.38	
Age (years)		Yes	54.5	24.5	0.732
		No	50.84	2.58	
ASA score		Yes	2.5	0.71	0.446
		No	2.12	0.67	
Pre-operative eGFR (mL/min/1.73 m^2^)		Yes	143.78	6.04	*0.001*
		No	75.65	4.84	
**Variable**	**Category**	**>25% reduction in eGFR**	**Frequency**	**Percentage**	** *p* ** **-value (Chi-Square)**
Sex	Male	Yes	0	0	0.127
		No	14	100	
	Female	Yes	2	15.38	
		No	11	84.62	
Inherited RCC syndromes	VHL	Yes	1	5.56	0.115
		No	17	94.44	
	HLRCC	Yes	0	0	
		No	1	100	
	HPCC	Yes	1	50.00	
		No	1	50.00	
	BHD	Yes	0	0	
		No	6	100	
Solitary kidney	Yes	Yes	0	0	0.603
		No	3	100	
	No	Yes	2	8.33	
		No	22	91.67	
Charlson Score	0	Yes	1	14.29	0.419
		No	6	85.71	
	>1	Yes	1	5.00	
		No	19	95.00	
Hypertension	Yes	Yes	2	22.22	*0.038*
		No	7	77.78	
	No	Yes	0	0.00	
		No	18	100.00	
Smoking	Yes	Yes	1	12.50	0.512
		No	7	87.50	
	No	Yes	1	5.26	
		No	18	94.74	
Obesity	BMI > 30	Yes	0	0.00	0.385
		No	7	100.00	
	BMI < 30	Yes	2	10.00	
		No	18	90.00	

## Data Availability

The data presented in this study are available in this article.

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
