# Peer review of "Oncological and Peri-Operative Outcomes of Percutaneous Cryoablation of Renal Cell Carcinoma for Patients with Hereditary RCC Diseases—An Analysis of European Multi-Centre Prospective EuRECA Registry"

_cancers, 2023, doi:10.3390/cancers15133322_

Round 1

Reviewer 1 Report

The paper describes the largest known cohort of patients with hereditary RCC underwent IGA/PCA. The paper is well written, interesting and well describes the management of these patients. I recommend the paper is accepted in the present form.

Author Response

Point 1: The paper describes the largest known cohort of patients with hereditary RCC underwent IGA/PCA. The paper is well written, interesting and well describes the management of these patients. I recommend the paper is accepted in the present form.

Response 1: Thank you very much for your kind comments.

Reviewer 2 Report

This large-scale study reports the oncological and peri-operative outcomes of PCA in patients with various hereditary RCC syndromes in high-volume centers across Europe, based on the EuRECA registry and demonstrates the importance of this minimally invasive approach to provide local oncological control without compromising renal function.

The study is clearly written and explains the importance of PCA in these patients, emphasizing the benefits of this interventional procedure.

Small changes need to be made to this work:

- clearly describe the types of CD complications.

- insert some images of the interventional procedures.

- visceral adipose tissue is a known risk factor for the development of clear cell renal cell carcinoma, linked to the pathogenesis (https://doi.org/10.4103/ccij.ccij_62_18). I would suggest, at the end of the discussion, to express this concept and to suggest a future study that can evaluate the amount of adipose tissue in patients affected by hereditary RCC disease, in order to better understand the pathogenesis and relationship of adipose tissue on these hereditary syndromes.

- further small changes:

In line 132 write the number 41 in full because it is the first word after the point.

In line 133 write the number 34% in full because it is the first word after the period.

Lines 142-144 it is necessary to write the number of patients with the Roman numeral and not with the word, so that it is coherent with the rest of the text.

In line 190 write number 5 of 5-year in full because it is the first word after the period.

In line 193 write the number 1 of 1-year in full because it is the first word after the period. Read the work so that the text is correct in this form.

Author Response

Point 1: This large-scale study reports the oncological and peri-operative outcomes of PCA in patients with various hereditary RCC syndromes in high-volume centers across Europe, based on the EuRECA registry and demonstrates the importance of this minimally invasive approach to provide local oncological control without compromising renal function.

Response 1: Thank you for your positive comments. 

Point 2: The study is clearly written and explains the importance of PCA in these patients, emphasizing the benefits of this interventional procedure.

Response 2: Thank you for your positive comments

Point 3: Clearly describe the types of CD complications

Response 3: Thank you for your suggestion. Clavien Dindo grade is now further described in the methods section (Line 119-122). 

Point 4: Insert some images of the interventional procedures.

Response 4: Thank you for the suggestion, however, images of the interventional procedures are not routinely collected from the registry hence we are not able to insert images of the interventional procedure. 

Point 5: Visceral adipose tissue is a known risk factor for the development of clear cell renal cell carcinoma, linked to the pathogenesis (https://doi.org/10.4103/ccij.ccij_62_18). I would suggest, at the end of the discussion, to express this concept and to suggest a future study that can evaluate the amount of adipose tissue in patients affected by hereditary RCC disease, in order to better understand the pathogenesis and relationship of adipose tissue on these hereditary syndromes.

Response 5: Thank you for your suggestion. This has now been added and discussed in discussion. 

Points 6-10: 

In line 132 write the number 41 in full because it is the first word after the point.

In line 133 write the number 34% in full because it is the first word after the period.

Lines 142-144 it is necessary to write the number of patients with the Roman numeral and not with the word, so that it is coherent with the rest of the text.

In line 190 write number 5 of 5-year in full because it is the first word after the period.

In line 193 write the number 1 of 1-year in full because it is the first word after the period. Read the work so that the text is correct in this form.

Response 6-10: Thank you for the comments and these have been addressed. 

Reviewer 3 Report

Treatment of hereditary renal cell carcinoma is challenging. I agree that to keep renal function is very important.

This manuscript might be valuable in near future.

However, the follow up period is not enough to discuss the survival.

I would like to know the location of tumor and distribution of tumor size. It might be affect oncological outcome.

How many tumors do you treat in one session?

How long has it been since previously treatment?

I would like to know the detail of patient characteristics of the case who had local recurrence.

(For example: HLRCC? Tumor size, tumor location, multiple or single, Solitary kidney)

Author Response

Point 1: Treatment of hereditary renal cell carcinoma is challenging. I agree that to keep renal function is very important.

This manuscript might be valuable in near future.

Response 1: Thank you very much for your positive comments. 

Point 2: However, the follow up period is not enough to discuss the survival.

Response 2: Thank you for the comments. The mean follow up period for the study is 30.4 months, with a standard deviation of 22 months, suggesting a proportion of patient would have received over 36 months of follow-up. While 3 years may be a short follow-up period to discuss survival in patients with sporadic RCC, in patients with hereditary disease, given the high rate of de novo disease, 3 years of follow-up would still provide valuable insight into the effectiveness of the treatment especially in terms of overall survival and cancer-specific survival. Furthermore, providing the rare nature of the disease and the difficulty to conduct further prospective studies and followup in these patients, 3 years/ 5 years outcomes are an important measure in these group of patients. 

Point 3: I would like to know the location of tumor and distribution of tumor size. It might be affect oncological outcome.

Response 3: Thank you for your comments. The location of the tumour is taken account into by the RENAL nephrometry score. It is a good predictor of treatment efficacy - Usefulness of R.E.N.A.L. nephrometry scoring system for predicting outcomes and complications of percutaneous ablation of 751 renal tumors - PubMed (nih.gov)

Unfortunately it is impossible to outline the location of the tumour without summarising with a measure such as the RENAL nephrometry score. The tumour size and RENAL nephrometry score is also summarised in table 1. 

Point 4: How many tumors do you treat in one session?

Response 4: Thank you for your comment. As this is a registry study, the number of tumour treated per session is down to the individual interventional radiologist's decision. However, as mentioned in line 154 amd 155, the mean number of RCCs treated per patient was 1.6. 

Point 5: How long has it been since previously treatment?

Response 5: Thank you for your comment. Unfortunately once again due to this being a registry study there is no information for the duration of redevelopment of denovo tumour as this falls outside the remit of the registry. 

Point 6: I would like to know the detail of patient characteristics of the case who had local recurrence.

(For example: HLRCC? Tumor size, tumor location, multiple or single, Solitary kidney)

Response 6: Thank you for the comment. This is now added in line 199-204. The recurrence was observed in a VHL's patient’s solitary left kidney (previous nephrectomy of their right kidney done due to RCC), and had a RENAL Nephrometry score of 9. The solid tumour measured 33mm in diameter and was < 50% exophytic. The anteriorly located lesion crossed the polar line for more than 50%, was less than 4mm from the collecting system, and touched the renal artery.

Reviewer 4 Report

Just a brief comment: the technique used in this research is cryoablation.

In many cdenters RFA would (still) be used, even for larger lesions.

There is a recent interesting publication on cancer-specific mortalitiy after cryoablation vs heat-based thermal ablation in T1a renal cell carcinoma. G. Sorce et all. JUrol vol 209, 81-88, Januari 2023

Author Response

Point 1: Just a brief comment: the technique used in this research is cryoablation. In many cdenters RFA would (still) be used, even for larger lesions.

Response 1: Thank you for your comment. There are debates between the optimal technique for small renal mass ablation and we agree this should be explored elsewhere especially with sporadic tumours and T1b tumours alongside newer modalities such as microwave and IRE. 

Point 2: There is a recent interesting publication on cancer-specific mortalitiy after cryoablation vs heat-based thermal ablation in T1a renal cell carcinoma. G. Sorce et all. JUrol vol 209, 81-88, Januari 2023

Response 2: Thank you we have now commented on this and cited the suggested manuscript. 

Round 2

Reviewer 3 Report

This study aims to evaluate the safety, efficacy and renal function preservation of percutaneous cryoablation (PCA) for small renal masses (SRMs) in inherited RCC syndromes. To preserve renal function, PCA is superior to partial nephrectomy(PN) and it is less invasive than PN.

Number of cases are small,, but ithis manuscript is useful information for treatment of inherited RCC.